# Phototrophy and carbon fixation in Chlorobi postdate the rise of oxygen

**L. M. Ward** [1,2]*, **Patrick M. Shih** [3,4,5,6]*

**1** Department of Earth and Planetary Sciences, Harvard University, Cambridge, Massachusetts, United States of America, **2** Department of Geosciences, Smith College, Northampton, Massachusetts, United States of America, **3** Department of Plant and Microbial Biology, University of California, Berkeley, Berkeley, California, United States of America, **4** Feedstocks Division, Joint BioEnergy Institute, Emeryville, California, United States of America, **5** Environmental Genomics and Systems Biology Division, Lawrence Berkeley National Laboratory, Berkeley, California, United States of America, **6** Innovative Genomics Institute, University of California, Berkeley, Berkeley, California, United States of America

* lward15@smith.edu (LMW); pmshih@berkeley.edu (PMS)

## Abstract

While most productivity on the surface of the Earth today is fueled by oxygenic photosynthesis, for much of Earth history it is thought that anoxygenic photosynthesis—using compounds like ferrous iron or sulfide as electron donors—drove most global carbon fixation. Anoxygenic photosynthesis is still performed by diverse bacteria in niche environments today. Of these, the Chlorobi (formerly green sulfur bacteria) are often interpreted as being particularly ancient and are frequently proposed to have fueled the biosphere during late Archean and early Paleoproterozoic time before the rise of oxygenic photosynthesis. Here, we perform comparative genomic, phylogenetic, and molecular clock analyses to determine the antiquity of the Chlorobi and their characteristic phenotypes. We show that contrary to common assumptions, the Chlorobi clade is relatively young, with anoxygenic phototrophy, carbon fixation via the rTCA pathway, and iron oxidation all significantly postdating the rise of oxygen ~2.3 billion years ago. The Chlorobi therefore could not have fueled the Archean biosphere, but instead represent a relatively young radiation of organisms which likely acquired the capacity for anoxygenic photosynthesis and other traits via horizontal gene transfer sometime after the evolution of oxygenic Cyanobacteria.

## Introduction

The Chlorobi (formerly Green Sulfur Bacteria) are a clade of anoxygenic phototrophic bacteria, classically known as strictly anaerobic sulfide oxidizers capable of carbon fixation via the rTCA pathway (e.g. [1]) but which has more recently been recognized as also including phototrophic iron oxidizers (photoferrotrophs) (e.g. [2]) and members that are aerobic and/or photoheterotrophic (e.g. [3, 4]). The Chlorobi were among the first described anoxygenic phototrophs over a century ago [5, 6] and have often been interpreted as being among the first phototrophs to evolve (e.g. [7]). The rTCA pathway used for carbon fixation in many Chlorobi is thought to be one of—if not the most–ancient carbon fixation pathway to have evolved (e.g [8, 9]). Subsequently, the discovery of iron oxidizing phototrophy (i.e. photoferrotrophy) in

**Data Availability Statement:** All data used in this study are derived from publicly available databases as described in the methods. NCBI Genbank accession numbers or WGS IDs as appropriate for all genomes used in this study are listed in S2 Table.

**Funding:** LMW was supported by an Agouron Institute Postdoctoral Fellowship and a Simons Foundation Postdoctoral Fellowship in Marine Microbial Ecology. The funders had no role in study design, data collection and analysis, decision to publish, or preparation of the manuscript.

**Competing interests:** The authors have declared that no competing interests exist.

some members of the Chlorobi has implicated this clade in fueling primary productivity and producing Banded Iron Formations during Precambrian time (e.g. [10]) when reduced iron was a much more abundant electron donor in the open oceans than sulfur (e.g. [11]). However, with the possible exception of aromatic carotenoid biomarkers in Mesoproterozoic strata (e.g. [12]), there is little to no rock record evidence for the antiquity of the Chlorobi [13]. Constraining the age and origin of Chlorobi, along with their characteristic metabolic traits—particularly anoxygenic phototrophy, carbon fixation via the rTCA pathway, and phototropic iron oxidation (photoferrotrophy)—is therefore an important open question for understanding the applicability of the physiology and ecology of extant Chlorobi as analogs for Archean and Paleoproterozoic primary producers. Here, we integrate comparative genomic, phylogenetic, and molecular clock techniques to determine the antiquity of the Chlorobi and their key metabolic pathways.

## Methods

Genomes of relevant organisms (including Chlorobi, other Bacteroidota, and outgroups) were downloaded from the NCBI Genbank and WGS databases, including genome data from [3, 4, 14–16]. Completeness and contamination statistics of metagenome-assembled genomes (MAGs) was estimated by CheckM [17] based on presence and copy number of conserved single-copy proteins. Sequences of proteins used in analyses were identified locally with the *tblastn* function of BLAST [18], aligned with MUSCLE [19], and manually curated in Jalview [20]. BLAST hits were retained if they were approximately full length (e.g. >90% the shortest reference sequence from an isolate genome) and had *e*-values better than $1e^{-20}$. The presence or absence of metabolic pathways of interest in MAGs was predicted with MetaPOAP [21]. Phylogenetic trees were calculated with using RAxML [22] on the Cipres science gateway [23]. Branch support values were calculated via transfer bootstrap support (TBE) by BOOSTER [24]. Trees were visualized with the ItoL viewer [25]. Taxonomic assignment was confirmed with GTDB-Tk [26–28]. Histories of vertical versus horizontal inheritance of metabolic proteins was made by comparison of organismal and metabolic protein phylogenies as described previously [29–30] Molecular clock analyses were performed following methods described briefly below, as established previously in [31, 32]. A concatenated protein sequence alignment was generated by extracting and aligning protein sequences for marker genes followed by concatenation of aligned sequences. Concatenated alignments were curated with Gblocks [33] and then manually in Jalview [20]. Taxa included in this alignment consist of all available Chlorobi genomes on the NCBI GenBank and WGS databases as well as sister groups and outgroups spanning the full diversity of the Bacteroidota and members of closely related phyla (e.g. Calditrichaeota and Zixibacterota) as assessed by GTDB [26–28] and concatenated ribosomal protein phylogenies of the tree of life [34], as well as Cyanobacteria, plastids, Proteobacteria, and mitochondria necessary for calibrating the molecular clock. Phylogenetic markers were chosen as conserved proteins across bacteria, plastids, and mitochondria, as previously reported [31, 32, 35], and consisted of AtpA, AtpB, EfTu, AtpE, AtpF, AtpH, AtpI, Rpl2, Rpl16, Rps3, and Rps12. Bayesian molecular clock analyses were carried out using BEAST v2.4.5 [36] on the Cipres science gateway [23]. As previously reported, the CpREV model was chosen as the best-fitting amino acid substitution model for the concatenated protein dataset based on ProtTest analysis [35]. Time constraints for the most recent common ancestor of Angiosperms (normal distribution with a mean of 217 Ma and sigma of 40 Ma) and land plants (normal distribution with a mean of 477 Ma and sigma of 70 Ma) were used as priors, leveraging cross-calibration techniques enabled by plastid and mitochondrial endosymbiosis events as described previously [37]. The most recent common ancestor of Rhodophytes was estimated via recently improved

estimates for the age of the fossil constraint *Bangiomorpha pubescens* [38]; we set this constraint as a uniform prior from 1030–1200 Ma to account for both Re-Os and Pb-Pb isotopic measurement estimates fir the age of *Bangiomorpha* [39]. A conservative uniform prior between 2300–3800 Ma was set on the divergence between oxygenic Cyanobacteria and Melainabacteria, as oxygenic photosynthesis evolved prior to the Great Oxygenation Event and most likely evolved sometime after the Late Heavy Bombardment. Finally, a uniform prior for all taxa was again set conservatively between 2400–3800 Ma, assuming that the Last Bacterial Common Ancestor most likely evolved after the Late Heavy Bombardment. Wide uniform priors were used to provide very conservative upper and lower limits. Three Markov chain Monte Carlo chains were run for 100 million generations sampling every 10,000th generation, and the first 50% of generations were discarded as burn-in. TreeAnnotator v1.7.5 [36] was used to generate maximum clade credibility trees.

## Results and discussion

### Chlorobi taxonomy

Despite classically being regarded as a separate phylum [40], our analyses robustly place the Chlorobi as a clade within the broader Bacteroidota (formerly Bacteroidetes) phylum (Fig 1). The phylogenetic relationships recovered here are broadly consistent with those determined by GTDB, whose taxonomic assignments we refer to throughout. This taxonomic revision is

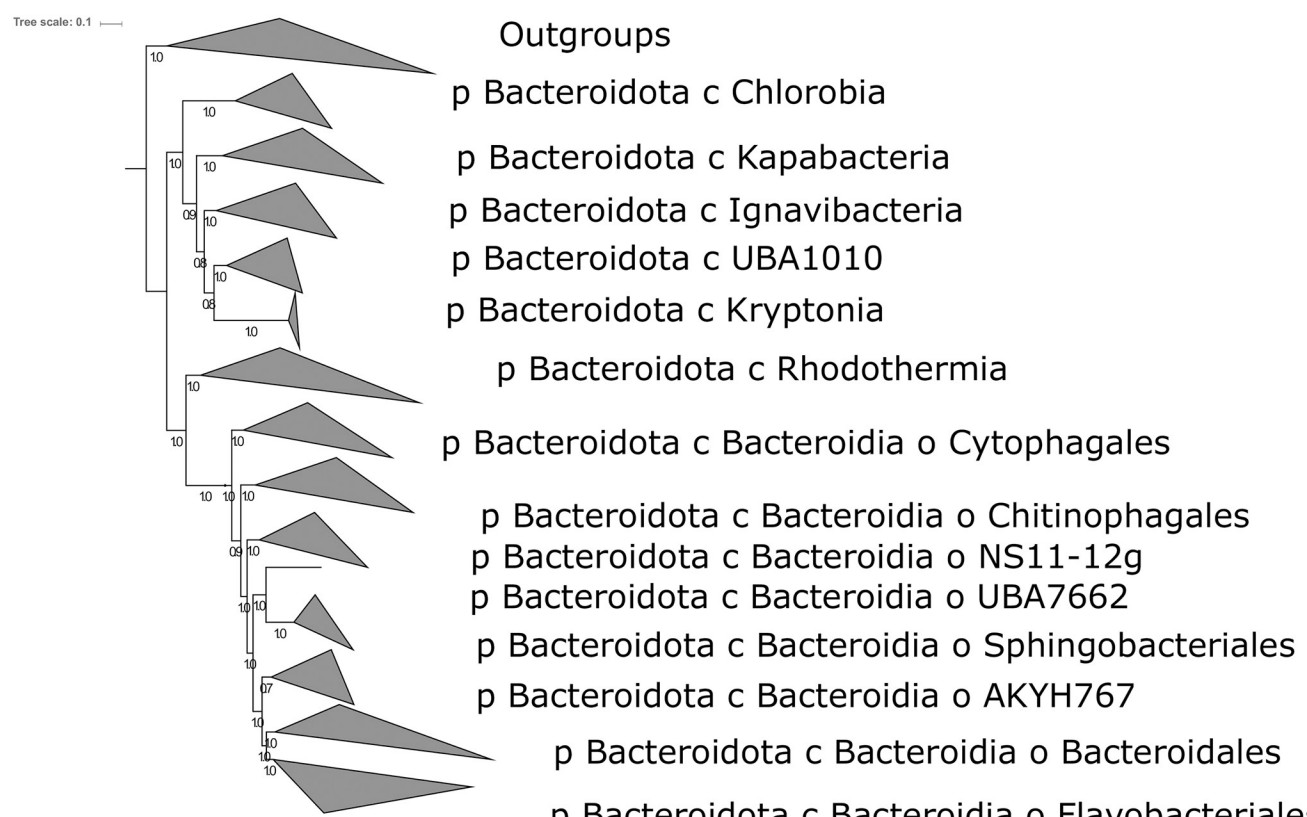

**Fig 1. Phylogeny of the Bacteroidota built with concatenated ribosomal proteins and labeled with clade names (classes, and orders within Bacteroidia) derived from GTDB-Tk.** The Chlorobia (formerly Chlorobi) are positioned as a class within the broader Bacteroidota phylum, and not as a separate phylum as previously thought.

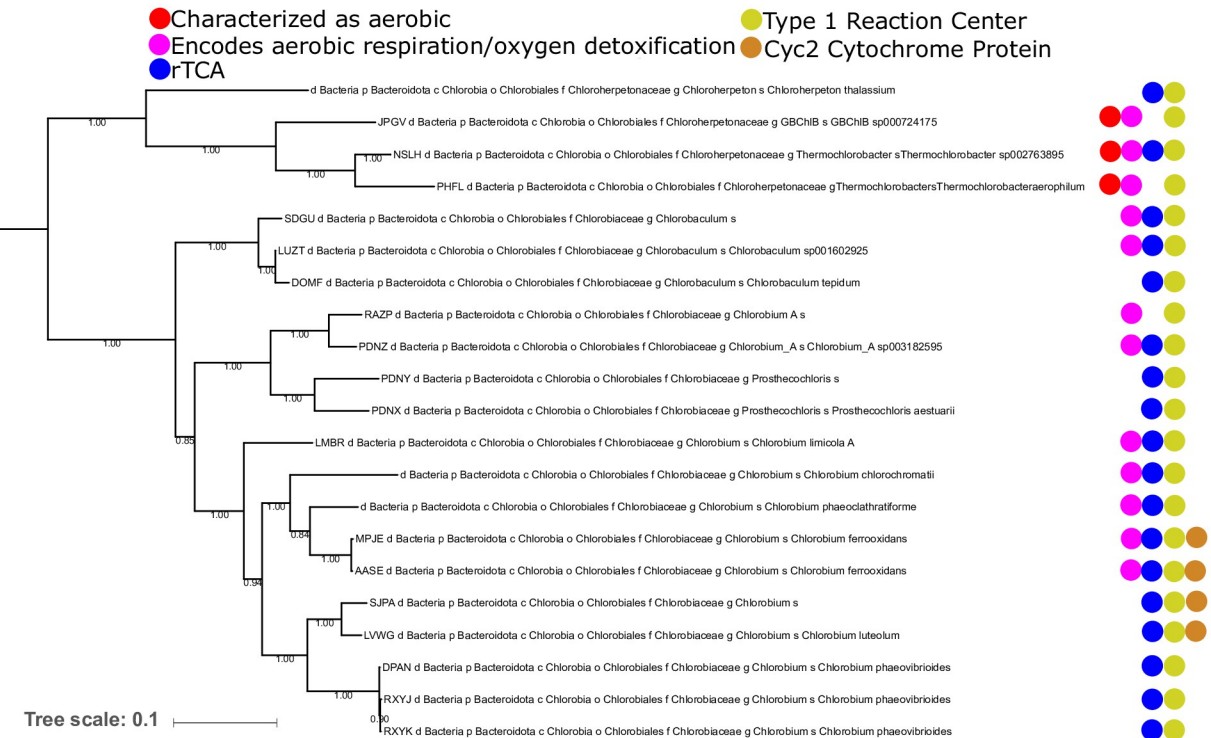

**Fig 2. Phylogeny of the Chlorobia built with concatenated ribosomal proteins, with taxonomic assignments from GTDB-Tk and the presence of metabolic traits discussed in the text mapped on.** The presence of a Type 1 phototrophic reaction center is ubiquitous within the Chlorobia class, but other traits (such as the presence of a Cyc2 cytochrome protein associated with iron oxidation) have a more limited phylogenetic distribution and may be more derived traits that were acquired after the radiation of the phylum.

based partially on improved deep branch support thanks to larger concatenated protein phylogenies versus older phylogenies based only on 16S rRNA or other single markers (iterating on improvements such as those described by [41]) as well as improved sampling of uncultured deep branches. In particular, it appears that the Chlorobi, together with a larger sister clade consisting of the classes Kapabacteria, Ignavibacteria, Kryptonia, and a class designated by GTDB as UBA10030, are basal within the broader Bacteroidota phylum. Phototrophy within the phylum appears to be confined to the Chlorobi class, consistent with other phyla where basal lineages are nonphototrophic and phototrophy appears to be a derived trait (e.g. [42]).

Evolutionary relationships within the Chlorobi are largely well resolved. Within the Chlorobi, all known genomes are restricted to a single order-level clade (Chlorobiales), consisting of two families as defined by GTDB and supported by our ribosomal phylogenies (Fig 2): Chlorobiaceae, the classically anaerobic photoautotrophic "green sulfur bacteria", and Chloroherpetonaceae, which contains putatively aerobic and photoheterotrophic members. While the relationships between and within these families is largely well resolved and robust to analysis with different methods and marker sets, an exception exists in the placement of *Chloroherpeton thalassium*, which varies between marker sets (discussed below). Other than *Chloroherpeton*, the Chloroherpetonaceae family includes two uncultured genera, *CBChlB* and *Thermochlorobacter*, which both include at least some putatively aerobic photoheterotrophic members. The Chlorobiaceae family includes three genera recognized by GTDB: *Chlorobium*, *Chlorobaculum*, and *Prosthecochloris* (although species previously classified as *Chlorobium* appear to be polyphyletic and has accordingly been divided into

*Chlorobium sensu stricto* and a second genus currently listed as *Chlorobium A*). Other genera previously proposed within the Chlorobi (e.g. *Clathrochloris*, *Ancalochloris*, and *Pelodictyon)* either have insufficient data (i.e. no type strain and no available sequence data for *Ancalochloris*, [1]) or have been reassigned to other genera (e.g. *Clathrochloris* and *Pelodictyon*) ([1, 26]).

## Distribution and evolutionary history of metabolic traits in the Chlorobi

Comparison of organismal relationships in the Chlorobi with metabolic protein distributions and phylogenies reveals a complex history of synapomorphic, vertically inherited, and laterally acquired genes responsible for several notable metabolic traits.

Phototrophy via a Type 1 Reaction Center appears to be a synapomorphy of crown group Chlorobi (i.e., the clade consisting of the last common ancestor of extant Chlorobi and all of its descendents), having been acquired in the stem lineage of the class after the divergence from other Bacteroidota. Consistent with this scenario, Type 1 Reaction Centers are absent in other Bacteroidota. Type 1 Reaction Centers in the Chlorobi form a monophyletic clade sister to the reaction centers of *Chloracidobacterium*; the distant organismal relationships of these lineages (in the Bacteroidota versus the Acidobacteria, phyla separated by many nonphototrophic lineages and several groups encoding Type 2 Reaction Centers, e.g. [13]) is consistent with these organisms acquiring genes for phototrophy via horizontal gene transfer from an unknown (possibly extinct) donor. However, as previously observed (e.g. [11, 43–45]), phylogenetic relationships of bacteriochlorophyll synthesis proteins are incongruent with relationships between reaction centers: the bacteriochlorophyll synthesis genes encoded by Chlorobi are most closely related to those of Chloroflexi [46]. This likely reflects independent histories of HGT in genes encoding different components of the phototrophy pathway, with Chlorobi acquiring bacteriochlorophyll synthesis from a different donor than reaction centers (potentially associated with the loss and replacement of older bacteriochlorophyll synthesis genes during acquisition of chlorosomes, e.g. [47, 48]).

The rTCA cycle for carbon fixation is essentially ubiquitous in the Chlorobiaceae family (with key genes absent only from a single metagenome-assembled genome of fairly low completeness, which is therefore likely a false negative result) and appears in some but not all members of the Chloroherpetonaceae (Fig 2). ATP Citrate Lyase, a key marker protein for the rTCA pathway (e.g. [49]), appears to be largely vertically inherited since its acquisition in the Chlorobi. The ATP Citrate Lyase encoded by the Chlorobi is most closely related to those found in nitrite oxidizing bacteria (NOB) in the Nitrospira and Nitrospina phyla (Fig 3); the apparent nesting of Chlorobi ATP Citrate Lyase within a larger clade primarily derived from NOB suggests that the Chlorobi acquired the capacity for the rTCA pathway via horizontal gene transfer from NOB. A history of HGT of genes associated with the rTCA cycle has been proposed previously (e.g. [50]), but to our knowledge the directionality and timing of these events and the evolution of the rTCA cycle as a whole has not been previously demonstrated (e.g. [13]). Our observation of the acquisition of the rTCA cycle by Chlorobi from NOB indirectly supports an origin for carbon fixation in Chlorobi after the origin of the aerobic nitrogen cycle ~2.3 billion years ago (e.g. [51]). The antiquity of rTCA in NOB relative to Chlorobi is further consistent with the higher sequence-level diversity of rTCA proteins such as ATP citrate lyase in NOB (>75% similarity between sequences from Chlorobi, >65% between Nitrospirota, as low as ~53% between Nitrospinota), consistent with NOB having radiated over substantially longer evolutionary timescales than rTCA-encoding Chlorobi. This is possibility is supported by molecular clock studies that have placed the radiation of Nitrospirota and Nitrospinota >1.5 Gya [31].

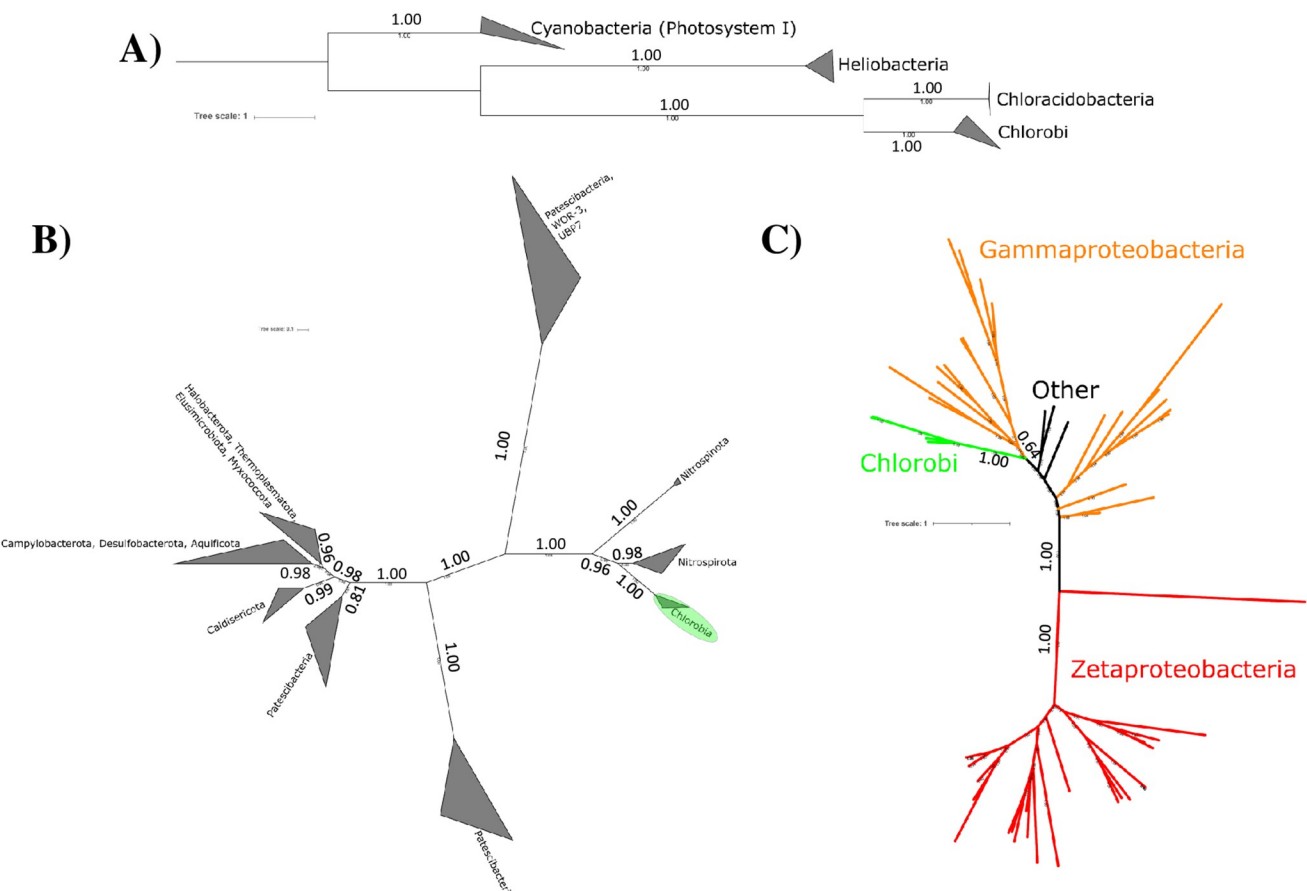

**Fig 3. Phylogenies of metabolic proteins discussed in the text; congruence of metabolic protein phylogenies with organismal phylogenies (e.g. those derived from concatenated ribosomal proteins such as in Fig 2) suggests vertical inheritance of traits from the last common ancestor, whereas incongruent topologies suggest a history of horizontal gene transfer of metabolic pathways.** A) Phylogeny of type 1 reaction centers, showing broad topological congruence with organismal relationships shown in Fig 2; together with the synapomorphic distribution of phototrophy in the Chlorobia, this suggests that phototrophy was acquired in stem group Chlorobia and has since been vertically inherited. B) Phylogeny of ATP citrate lyase, a key marker gene for the capacity for carbon fixation via the rTCA pathway. Phylogenetic relationships of this gene within the Chlorobia are congruent with organismal relationships, suggesting that this trait has been vertically inherited in most Chlorobi (with the exception of loss in aerobic members of the Chloroherpetonaceae). However, ATP citrate lyase in Chlorobi appears to be derived from genes of nitrite oxidizing bacteria (Nitrospirota and Nitrospinota), suggesting that this trait was acquired by Chlorobi via horizontal gene transfer after the radiation of nitrite oxidizers. C) Phylogeny of Cyc2 cytochrome proteins associated with iron oxidation in Chlorobi and Zetaproteobacteria. Cyc2 genes in the Chlorobi appear to be derived from those of Proteobacteria, suggesting that the capacity for iron oxidation was acquired by Chlorobi via horizontal gene transfer after the radiation of aerobic iron oxidizing bacteria.

As both phototrophy and carbon fixation appear to be derived traits within total group Chlorobi, acquired after the divergence of this class from other lineages of Bacteroidota, the maximum and minimum ages of total group Chlorobi can be used to determine the age of photosynthesis in this clade. This is simplified by evidence that these traits have been vertically inherited within the Chlorobi with little or no horizontal gene transfer after their first acquisition as shown by the broad congruence of organismal relationships with phylogenies of proteins involved in phototrophy and carbon fixation (e.g. PscA, ATP citrate lyase) (Figs 2 and 3). The initial acquisition of these traits in stem group Chlorobi is best explained by horizontal gene transfer from other phyla, however.

The capacity for iron oxidation in Chlorobi is restricted to a small number of lineages in the genus *Chlorobium*. This is most consistent with a relatively late, secondary acquisition of this

trait long after the radiation of the Chlorobi. Photoferrotrophic Chlorobi (but not non-photo-trophic close relatives) encode a Cyc2 Cytochrome protein homologous to that putatively used in iron oxidation by members of the Zetaproteobacteria such as *Mariprofundus ferrooxidans* [52], leading to the use of Cyc2 as a marker for photoferrotrophy in Chlorobi genomes although the function of this protein in iron oxidation by Chlorobi has not yet been tested in culture [53]. Phylogenetic relationships of Cyc2 Cytochrome proteins suggest that iron oxidizing Chlorobi may have acquired this trait via HGT from members of the Proteobacteria phylum. Phylogenetic placement of Cyc2 Cytochromes from Chlorobi are in a fairly derived position, sister to a clade made up of iron oxidizing Betaproteobacteria such as *Gallionella* and *Sideroxydans*, nested within a broader radiation of proteobacterial sequences. Recent molecular clock estimates for the antiquity of Zeta- and Beta-proteobacteria suggest that these clades are less than ~1.8 Ga, with crown group Proteobacteria ~2.2 Ga [31]; together with our results here, this suggests that much if not all of the diversity of neutrophilic iron oxidizing bacteria that utilize Cyc2 Cytochrome homologs have radiated after the Great Oxygenation Event ~2.3 Gya when aerobic metabolisms first became viable due to the accumulation of significant atmospheric $O_2$.

Depending on the marker set used (e.g. concatenated ribosomal proteins versus the broader marker sets used by GTDB-Tk and in the construction of our molecular clocks) the position of *Chloroherpeton thalassium* varies between a position sister to Chlorobiaceae and one basal to the aerobic clade of Chloroherptonaceae (e.g. Figs 2 and 4, [54]). This ambiguity in the proper phylogenetic placement of *C. thalassium* has substantial implications for interpretations of the antiquity of aerobic respiration in the Chlorobi—if *C. thalassium* clusters with the aerobic clade, this implies that the last common ancestor of the Chlorobi was anaerobic and that aerobic respiration was acquired later within the Chloroherpetonaceae, while if it branches between the two families it is equally parsimonious that the last common ancestor of the Chlorobi was aerobic but that this trait was secondarily lost in the Chlorobiaceae. While we are unable to confidently choose between these scenarios at this time, our analyses regarding the timing of other traits in Chlorobi are robust to this uncertainty.

## Molecular clock estimates for the antiquity of the Chlorobi

Molecular clock analyses are consistent with an origin of total group Chlorobi after the Great Oxygenation Event ~2.3 Gya (Fig 4, Table 1, S1 Table, S3–S8 Figs). The age of photoferrotrophy in the Chlorobi appears to be significantly younger, postdating the radiation of *Chlorobium* <300 Ma. The antiquity of aerobic respiration and the rTCA pathway in the Chlorobi depends on interpretations of where in the tree of Chlorobi these traits were first acquired (as discussed above), but in every scenario are restricted to total group Chlorobi with an estimated age of <2.0 Ga. Together with phylogenetic data described above, these data suggest that the Chlorobi acquired the capacity for anoxygenic photosynthesis after the evolution of oxygenic photosynthesis in Cyanobacteria led to the Great Oxygenation Event ~2.3 Gya. The evolution of photoferrotrophy in the Chlorobi appears to be a particularly evolutionarily derived trait that evolved much more recently in Earth history than has been previously been suggested (e.g., [7, 10]).

In contrast to previous molecular clock studies, our analyses place the total group (including stem group) Chlorobi after the GOE. Previous studies utilizing horizontal gene transfer events to help constrain molecular clocks also recapitulate crown group Chlorobi postdating the rise of oxygen in the atmosphere [55, 56]; however, these studies argue that stem lineages can predate the GOE. A major difference between our study and those performed previously is the sampling of nonphototrophic outgroups to the Chlorobi. The work of Magnabosco et al.

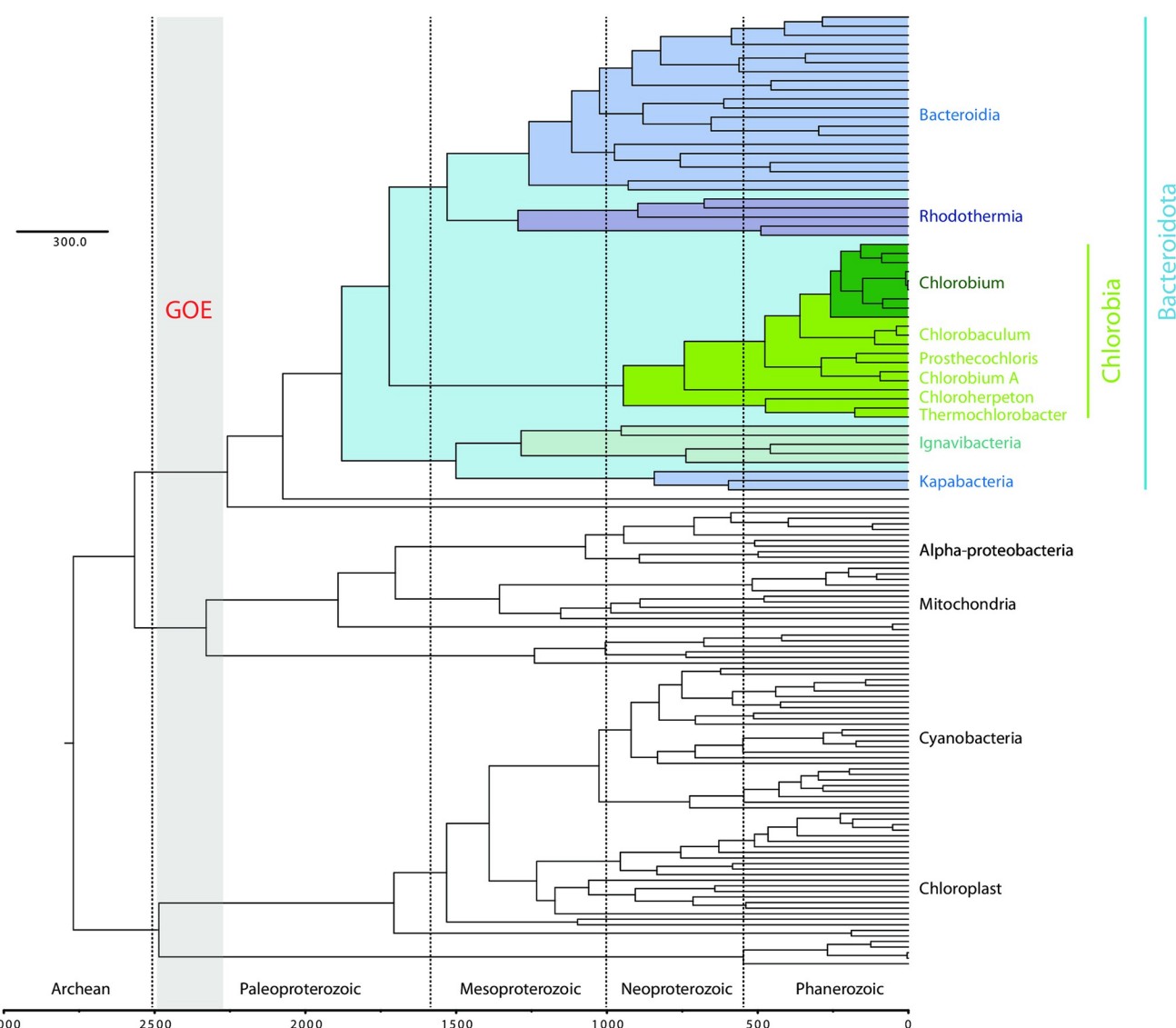

**Fig 4. Best-estimate molecular clock estimate showing age of Chlorobi.** Node ages based on T03 constraints (omitting biomarkers) (Table 1, S1 Table, S3 and S4 Figs). The Bacteroidota phylum (highlighted in blue) is estimated to have radiated well after the Great Oxygenation Event ~2.3 Gya (vertical gray bar). Crown group Chlorobi (highlighted in light green) originated in Neoproterozoic time <1 Gya. All known photoferrotrophic Chlorobi are found in the *Chlorobium* genus (highlighted in dark green) which originated <300 Mya.

**Table 1. Molecular Clock estimates for key clades based on T03 constraints.**

| Clade: | Notable traits constrained | Node Age (Ma) | 95% Confidence Interval (Ma) |
|---|---|---|---|
| Total Group Chlorobi | Maximum age of phototrophy and carbon fixation | 1880 | 1440–2030 |
| Crown Group Chlorobi | Minimum age for phototrophy and carbon fixation | 945 | 700–1236 |
| Total Group Chlorobiaceae | Aromatic carotenoid biomarkers | 744 | 536–980 |
| Crown Group *Chlorobium* | Iron oxidation/Photoferrotrophy | 259 | 184–334 |

[55] and Daye et al. [56] utilized only a small number of Ignavibacteria as outgroups to the Chlorobi. Our study includes a much larger sampling of the known diversity of Bacteroidota, revealing that Chlorobia branches as a sister group to Bacteroidia+Rhodothermia with Ignavibacteria as part of a deeper branching clade. As a result, any age estimate for total group Chlorobi with its origin taken as the Chlorobia/Ignavibacteria split will necessarily result in an overestimate in the potential age for phototrophy in this group due to artificial lengthening of the stem lineage as compared to our work that shows that phototrophy must postdate the younger Chlorobia/Bacteroidia split. This highlights the necessity for adequate sampling of outgroups and the breaking up of artificially long stem lineages for using phylogenetic and molecular clock approaches for determining the antiquity of derived traits. Even with our improved sampling of Bacteroidota, we still observe a long Chlorobi stem lineage (representing ~900 Ma of evolution). This is particularly problematic given that in the absence of additional information it is impossible to pinpoint when a trait like phototrophy was first acquired along a stem lineage—the acquisition of phototrophy is equally likely to have occurred immediately prior to the radiation of the crown group as it is to have occurred near the base of the total group.

## Theantiquity of Chlorobi and the rock record

Ideally, estimates for the antiquity of organisms and metabolisms derived from molecular clocks would be directly tested in the rock record. Unfortunately, phototrophic Chlorobi and carbon fixation via the rTCA cycle have left no direct evidence in the rock record [13]. The best evidence for these processes is indirect via the presumed link between phototrophic Chlorobi and lipid biomarkers such as isorenieratane and chlorobactene which have been recovered from ~1.64 Ga rocks in the Barney Creek formation [12]. These compounds are frequently considered diagnostic for members of the Chlorobiaceae as these organisms are typically considered to be the only producers of the biological precursors of these biomarker compounds (isorenieratene and chlorobactene, respectively) (e.g. [57]). However, the study of putatively diagnostic organic biomarkers has frequently revealed the production of identical compounds by diverse microbes via both convergent evolution (e.g. convergent evolution of 24-isopropyl cholesterol in algae and sponges, [58]) and horizontal gene transfer (e.g. origin of hopanoids in Proteobacteria and subsequent transfer to Cyanobacteria and other lineages, [59]). Recent work has shown that aromatic carotenoids can be produced by oxygenic photosynthetic Cyanobacteria [60], and that specific biomarkers that may be more diagnostic for Chlorobiaceae did not become prominent in the rock record until Phanerozoic time [60]. Even today putative Chlorobi biomarkers are known to be produced by nonphototrophic bacteria including some Actinobacteria (e.g. [61]) and it is challenging to state with certainty that other organisms have not produced these compounds at other times in Earth history. It is therefore not *a priori* obvious whether putative Chlorobi biomarkers provide a helpful or necessary molecular clock calibrations.

In order to test the impact of Barney Creek biomarkers on our molecular clocks, we performed additional analyses which used 1.64 Ga as a minimum age for crown group Chlorobi or as a minimum age for crown group Chlorobiaceae. Compared to analyses which omitted the biomarker constraints, these analyses resulted in older ages for crown group Chlorobi but also introduced much larger 95% confidence intervals for all nodes (with the exception of nodes directly constrained by biomarkers, which clustered tightly at the constraint boundary) (S3–S8 Figs). We therefore prefer the analyses which omit the biomarker constraints that result in the narrowest confidence intervals, but note that our conclusions are robust to the inclusion of biomarker constraints.

All three of our tested scenarios produce results that suggest that crown group Chlorobi postdate the GOE and that photoferrotrophy in the Chlorobi postdates the Precambrian/Cambrian boundary (S3–S8 Figs); the age of crown group Chlorobi and implications for the producers of 1.64 Ga biomarkers differ, however. Under our preferred analysis which omits biomarker calibrations, stem but not crown group Chlorobi were present by 1.64 Ga. This suggests that synthesis of isorenieratene and chlorobactene first evolved in stem group Chlorobi or another lineage which was responsible for producing these compounds by 1.64 Ga. The distribution of biomarker synthesis in extant Chlorobi (S1 Fig) is therefore best explained by presence of chlorobactene and isorenieratene production in the last Chlorobi common ancestor followed by loss of one or both in some lineages or, alternatively, horizontal gene transfer of the capacity for biomarker synthesis into the Chlorobiaceae from an earlier, unknown, biomarker producer which was responsible for the production of these compounds 1.64 Ga. The former scenario would be supported by the discovery of biomarker production in the Chloroherpetonaceae, but to our knowledge the biomarker synthesis capacity of Chloroherpetonaceae has not been determined with the exception of *Chloroherpeton thalassium*. It therefore remains possible that biomarker capacity is found throughout crown group Chlorobi, which would be consistent with the evolution of this trait in stem group Chlorobi by 1.64 Ga.

We can better distinguish scenarios for the antiquity of Chlorobi biomarkers by assessing the role of horizontal gene transfer in driving the distribution of biomarker synthesis. Ancient horizontal gene transfer can be identified via incongruity in organismal and functional protein phylogenies (e.g. [29]). A useful marker for biomarker synthesis in Chlorobi is CrtU, a protein involved in desaturation and isomerization of gamma carotene to produce chlorobactene and/or isorenieratene [43]. Phylogenies of CrtU appear to be broadly congruent with organismal phylogenies within the Chlorobiaceae (Fig 2, S2 Fig), suggesting vertical inheritance from the last common ancestor of this clade. However, the distribution of individual biomarkers is scattered (i.e., chlorobactene and isorenieratene producers are intermixed in the Chlorobi phylogeny, S1 Fig). Two scenarios are consistent with this distribution. Either the last common ancestor of the Chlorobiaceae was capable of producing both isorenieratene and Chlorobactene, followed by loss of one or the other compound in most lineages, or multiple evolutionary transitions have occurred between predominantly isorenieratene synthesis or predominantly chlorobactene synthesis in different Chlorobi strains. The latter instance in particular would not be consistent with biomarker records providing a phylogenetically constrained signal, as transition and reversion between biomarker products makes interpretation of the ancient producers of biomarker compounds difficult. Additionally, the presence of aromatic carotenoid synthesis in nonphototrophic organisms such as *Streptomyces griseus* [61] and the presence of closely related homologs of CrtU and other biomarker synthesis genes in taxa including an apparently nonphototrophic member of the uncultured GWC2-55-46 phylum (S2 Fig) suggest that the production of aromatic carotenoid biomarkers may be a more widely distributed and less taxonomically constrained process than previously recognized. We therefore remain open to the possibility that 1.64 Ga biomarkers of the Barney Creek formation may not record the presence of crown group Chlorobiaceae but may instead have been produced by stem group Chlorobi or other organisms and support the omission of biomarkers from our molecular clock calibrations.

## Conclusions

While the Chlorobi have often been interpreted as a particularly ancient group of phototrophs, potentially responsible for fueling the biosphere before the rise of oxygenic photosynthesis in Cyanobacteria (e.g. [10]), our results suggest that photosynthesis in the Chlorobi is instead a

relatively derived trait that significantly postdates the Great Oxygenation Event ~2.3 Gya. While Chlorobi may have been capable of fueling significant amounts of primary productivity during Meso- to-Neo-proterozoic time (e.g. [62]), these organisms likely had not yet radiated or acquired necessary traits to be able to drive primary productivity during Paleoproterozoic or Archean time. While metabolisms like RCI-driven phototrophy and rTCA-driven carbon fixation are likely very ancient, they did not originate in the Chlorobi but instead were acquired sometime during Proterozoic time via horizontal gene transfer from more ancient (and potentially now extinct) lineages.

Our results suggest that Chlorobi did not contribute to the deposition of Archean or Paleoproterozoic Iron Formations; instead, it appears that photoferrotrophy is a derived trait acquired in diverse bacterial lineages through a variety of processes including horizontal gene transfer and convergent evolution. In contrast, oxidation of reduced sulfur compounds is nearly ubiquitous in the Chlorobiaceae (e,g. [1]) and is likely a a synapomorphy of this clade acquired in stem lineages during Proterozoic time. It is likely that many lineages of photoferrotrophs have arisen throughout Earth history, many of which likely used nonhomologous proteins for iron oxidation; still today multiple mechanisms for iron oxidation are present between and even within bacterial phyla (e.g. [63]). As a result, care must be taken when extrapolating the physiology and ecology of ancient photo(ferro)trophs from that of extant organisms. While net metabolic processes may be similar throughout Earth history, the organisms themselves—as well as their fundamental biochemistry and their interactions with other organisms—may have been very different at times in Earth's past before extant clades evolved. Modern photo(ferro)trophs are not necessarily relicts of ancient diversity as previously presumed (e.g. [7]) but may instead reflect more recent radiations following horizontal gene transfer and/or convergent evolution of metabolic traits. An essential step in investigating extant photo(ferro)trophs in order to understand the ancient Earth is therefore to distinguish between traits that may be general to any organism performing a particular metabolism and those that may be lineage specific and which would vary in organisms utilizing different carbon fixation pathways, (bacterio)chlorophyll compounds, and iron oxidation mechanisms. Extant photoferrotrophs provide our only means to experimentally test how ancient photoferrotrophs may have made a living, but it's important to recognize that extant organisms may be palimpsests resulting from the accumulation of both ancient and recent evolutionary innovation and not direct vestiges of the organisms that performed these metabolisms deep in Earth history. It is therefore necessary to careful consider assumptions about and extrapolations from modern analogues before using them as the basis for claims about ancient organisms.

## Supporting information

**S1 Table. Molecular clock results for clades discussed in the text using different constraints (T03, T07, T09), based on S3–S8 Figs.** Node Age and 95% Confidence Intervals given in millions of years.
(TXT)

**S2 Table. NCBI Genbank/WGS IDs of all isolate genomes and metagenome-assembled genomes used in analyses.**
(TXT)

**S1 Fig. Distribution of aromatic carotenoid biomarkers in characterized Chlorobi strains, following Garcia Costas et al. 2012, Imhoff 2014.**
(PDF)

**S2 Fig. Protein phylogeny of CrtU, involved in the desaturation and isomerization of gamma carotene to produce chlorobactene and/or isorenieratene (Bryant et al., 2012).**
(PDF)

**S3 Fig. Molecular clock analysis of T03 cross-calibrated BEAST run displaying node age values.** Cyanobacteria/Melainabacteria divergence constrained to a uniform prior 2400–3800 Mya.
(PDF)

**S4 Fig. Molecular clock analysis of T03 cross-calibrated BEAST run displaying 95% Confidence Interval values.** Cyanobacteria/Melainabacteria divergence constrained to a uniform prior 2400–3800 Mya.
(PDF)

**S5 Fig. Molecular clock analysis of T07 cross-calibrated BEAST run displaying node age values.** Most recent common ancestor of Chlorobi constrained to uniform prior between 1637–1643 Mya. Cyanobacteria/Melainabacteria divergence constrained to a uniform prior 2400–3800 Mya.
(PDF)

**S6 Fig. Molecular clock analysis of T07 cross-calibrated BEAST run displaying 95% Confidence Interval values.** Most recent common ancestor of Chlorobi constrained to uniform prior between 1637–1643 Mya. Cyanobacteria/Melainabacteria divergence constrained to a uniform prior 2400–3800 Mya.
(PDF)

**S7 Fig. Molecular clock analysis of T09 cross-calibrated BEAST run displaying node age values.** Most recent common ancestor of Chlorobiaceae constrained to uniform prior between 1637–1643 Mya. Cyanobacteria/Melainabacteria divergence constrained to a uniform prior 2400–3800 Mya.
(PDF)

**S8 Fig. Molecular clock analysis of T09 cross-calibrated BEAST run displaying 95% Confidence Interval values.** Most recent common ancestor of Chlorobiaceae constrained to uniform prior between 1637–1643 Mya. Cyanobacteria/Melainabacteria divergence constrained to a uniform prior 2400–3800 Mya.
(PDF)

## Author Contributions

**Conceptualization:** L. M. Ward, Patrick M. Shih.

**Data curation:** L. M. Ward, Patrick M. Shih.

**Formal analysis:** L. M. Ward, Patrick M. Shih.

**Funding acquisition:** L. M. Ward.

**Investigation:** L. M. Ward, Patrick M. Shih.

**Methodology:** L. M. Ward, Patrick M. Shih.

**Project administration:** L. M. Ward, Patrick M. Shih.

**Resources:** L. M. Ward, Patrick M. Shih.

**Software:** L. M. Ward, Patrick M. Shih.

**Supervision:** L. M. Ward, Patrick M. Shih.

**Validation:** L. M. Ward, Patrick M. Shih.

**Visualization:** L. M. Ward, Patrick M. Shih.

**Writing – original draft:** L. M. Ward, Patrick M. Shih.

**Writing – review & editing:** L. M. Ward, Patrick M. Shih.

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
