## [Decision Letter · Decision Letter 0]

6 Apr 2022

PONE-D-21-25104

Phototrophy and carbon fixation in Chlorobi postdate the rise of oxygen

PLOS ONE

Dear Dr. Ward,

Thank you for submitting your manuscript to PLOS ONE. After careful consideration, we feel that it has merit but does not fully meet PLOS ONE’s publication criteria as it currently stands. Therefore, we invite you to submit a revised version of the manuscript that addresses the points raised during the review process.

Dear Dr Ward,

Thank you for your submission to Plos One. Two experts in the field have now reviewed your manuscript and they agree that it presents interesting new data and conclusions. Only minor revisions are required. Once those are completed, the manuscript should be ready for publication. 

Best wishes,

Eva Stueeken

We look forward to receiving your revised manuscript.

Kind regards,

Eva Elisabeth Stüeken, Ph.D.

Academic Editor

PLOS ONE

“LMW acknowledges support from an Agouron Institute Postdoctoral Fellowship, a Simons Foundation Postdoctoral Fellowship in Marine Microbial Ecology, and an NSF XSEDE Startup Award that provided computational resources via the CIPRES Science Gateway.”

“LMW was supported by an Agouron

Institute Postdoctoral Fellowship and a Simons

Foundation Postdoctoral Fellowship in Marine

Microbial Ecology. The funders had no role in study

design, data collection and analysis, decision to

publish, or preparation of the manuscript.”

3. We noted in your submission details that a portion of your manuscript may have been presented or published elsewhere. [DETAILS AS NEEDED] Please clarify whether this [conference proceeding or publication] was peer-reviewed and formally published. If this work was previously peer-reviewed and published, in the cover letter please provide the reason that this work does not constitute dual publication and should be included in the current manuscript.

Additional Editor Comments:

Dear Dr Ward,

Your manuscript has now been reviewed by two experts in the field. They find it compelling but identified a few minor points to address (see below). Please address these points in a revised version of the manuscript.

Best wishes,

Eva Stueeken

Reviewers' comments:

Reviewer's Responses to Questions

**Comments to the Author**

1. Is the manuscript technically sound, and do the data support the conclusions?

Reviewer #1: Yes

Reviewer #2: Yes

2. Has the statistical analysis been performed appropriately and rigorously? 

Reviewer #1: Yes

Reviewer #2: Yes

3. Have the authors made all data underlying the findings in their manuscript fully available?

Reviewer #1: Yes

Reviewer #2: Yes

4. Is the manuscript presented in an intelligible fashion and written in standard English?

Reviewer #1: Yes

Reviewer #2: Yes

5. Review Comments to the Author

Reviewer #1: The authors present data suggesting the emergence of Chlorobi occurred more recently in Earth’s history, perhaps in disagreement with others that green sulfur bacteria could have been early evolving phototrophs. The paper and results are generally straightforward. I have a few comments

The authors suggest they will discuss characteristic phenotypes but most of the conclusions are based on photoferrotrophy which (in genomes and isolates) is already a rare feature. Because a number of Chlorobi oxidize reduced sulfur (consistent with arguments that they could have been prevalent in the Archean), it seems relevant to also discuss this physiology (or to more explicitly justify the focus on photoferrotrophy).

A careful definition of “crown group” Chlorobi would be beneficial for the reader.

Provide refs: The evolution of photoferrotrophy in the Chlorobi appears to be a particularly

evolutionarily derived trait that evolved much more recently in Earth history than has been previously been suggested.

Figure 1 - the scale and bootstraps are impossible to read. Please increase the font size and resolution of the figure.

Figure 3 - same comment. This must have something to do with the PDF conversion in the PLoS system but I cannot read the labels on B. But the bootstrap font size needs to be larger.

Reviewer #2: Ward and Shih investigate the evolutionary history of phototrophy, carbon fixation, and iron oxidation in the chlorobi, a group of organisms commonly thought to have driven the anaerobic oxidation of iron in oceans on early Earth. The results of phylogenetic analyses, comparative genomic analyses, and molecular clock analyses lead the authors to conclude that the Chlorobi are not an ancient group of anoxygenic phototrophs and were unlikely to be responsible for the deposition of BIFs. Based on the data presented, I agree with the authors, and am happy to see the work come out in deference to several heated discussions I have had with colleagues that have argued the same thing, albeit with less strong evidence. I only have several edits/comments to suggest to the authors to further improve this paper.

Line numbers would be nice.

The abstract makes several vague claims about the role of anoxygenic photosynthesis in driving most global carbon fixation on “early earth”. This is true, but only during the latter parts of the archaeon. Prior to this, there would not have been phototrophy and it would have been chemotrophy as the primary driver. The authors need to add clarity to the time frames they are referencing. This is also true in the next sentence – do not discount the role of chemotropy – it does not hurt your arguments about the role of chlorobi in phototrophic production.

First paragraph of introduction. Most people regard the Wood Ljungdahl pathway as the most primitive CO2 fixation pathway (see papers by Martin, Russell, Boyd, and Shock). Those organisms that use the rTCA are primarily aerobes (at least those that are somewhat deeply rooted). This information was not known at the time Wachterhauser made his thoughts known.

Methods. Why not use the far more robust (albeit computationally more intensive) Clustal program to align sequences. Last line of this paragraph – remove non-essential self-citations.

Page 3 methods. The uniform prior of 2300-3800 for the split between Cynaobacteria and Melainabacteria is unnecessarily large. See the recent papers by Fournier for perhaps more reasonable calibration dates.

Page three, first paragraph results. The Chlorobi are a basal group among a clade that is sister to the other large clade. It is not necessarily sister to the other members within this clade. That it is basal is what is important to your argument.

Page 4. Are type I RC also absent in other Bacterioidota? That seems to be critical to your argument. Also, what are the type I RC in the Chlorobi most closely related to? Finally, the idea that the RC phylogeny and the Bch synthesis protein phylogenies are incongruent means that RCs may have evolved and then were augmented by chlorophylls? Is this common in other organisms as a means of energy generation? It would be good to speculate on how this could occur based on data that we have from other organisms where this is the case (aerobic anoxygenic phototrophs, for example). The implication is RCs can function without BChl?

Page 4 first paragraph. You don’t acquire phototrophy via HGT but you do acquire the genes that enable phototrophy via HGT.

Page 4 second paragraph. Your own data indicate that the rTCA cycle is not ubiquitous.

Page 4 second paragraph, second line to the end. This is a supporting argument at best. I would consider removing this as this phenomenon could equally be explained by expansion of niche space or variable rates of divergence/substitution.

Page 5, second paragraph. Have Cyc2 genes been shown to be involved in Fe oxidation in Chlorobi. Need to provide a citation or qualify as such. Also, the use of significant in the last sentence implies that you have performed a statistical test. Use a different word here.

Page 6, last line of first paragraph needs a reference to authors who have suggested chlorobi are old.

Conclusions. Have you ruled out photoferrotrophy in other extant lineages? Is it possible that photoferrotrophs abandoned iron when sulfide became more prevalent in oceans, etc. and thus we have no record of them. Please do a better job of trying to explain the presence of BIFs prior to the GOE here in a more ecologically appropriate manner (i.e., link changing Earth landscapes with changing selective pressures, etc.). Fe oxidation may only exist today in niche (rare) environments.

6. PLOS authors have the option to publish the peer review history of their article (what does this mean?). If published, this will include your full peer review and any attached files.

Reviewer #1: No

Reviewer #2: No

---

## [Author Response · Author response to Decision Letter 0]

13 May 2022

Many thanks to the editor and the two anonymous reviewers for their helpful feedback on our manuscript. We have now revised our manuscript and figures according to their recommendations. The revised manuscript is attached, along with a version utilizing the Track Changes feature of MS Word to highlight revisions. We are also uploading revised versions of our figures. Specific editor and reviewer comments are copied below, with our responses following.

3. We noted in your submission details that a portion of your manuscript may have been presented or published elsewhere. [DETAILS AS NEEDED] Please clarify whether this [conference proceeding or publication] was peer-reviewed and formally published. If this work was previously peer-reviewed and published, in the cover letter please provide the reason that this work does not constitute dual publication and should be included in the current manuscript.

Response: a preprint was made available at biorxiv (https://www.biorxiv.org/content/10.1101/2021.01.22.427768v1.abstract), but this manuscript has not been peer-reviewed and formally published elsewhere.

Response: we are happy for both authors to be listed as corresponding author, correcting this issue. 

Reviewers' comments:

Reviewer #1: The authors present data suggesting the emergence of Chlorobi occurred more recently in Earth’s history, perhaps in disagreement with others that green sulfur bacteria could have been early evolving phototrophs. The paper and results are generally straightforward. I have a few comments

The authors suggest they will discuss characteristic phenotypes but most of the conclusions are based on photoferrotrophy which (in genomes and isolates) is already a rare feature. Because a number of Chlorobi oxidize reduced sulfur (consistent with arguments that they could have been prevalent in the Archean), it seems relevant to also discuss this physiology (or to more explicitly justify the focus on photoferrotrophy).

Response: This is a fair point! We agree that sulfur oxidation is more widespread (and likely older) in Chlorobi than is photoferrotrophy. We assumed that sulfur oxidation would have been acquired along the same branch as photosynthesis, but we never said this explicitly. We focused on iron oxidation primarily because it was likely much more available in Archean oceans, and therefore is more relevant for whether Chlorobi may have driven primary productivity at this time. We have added to our intro and conclusions to make this more clear :

“Subsequently, the discovery of iron oxidizing phototrophy (i.e. photoferrotrophy) in some members of the Chlorobi has implicated this clade in fueling primary productivity and producing Banded Iron Formations during Precambrian time (e.g. Thompson et al. 2017) when reduced iron was a much more abundant electron donor in the open oceans than sulfur (e.g. Ward et al., 2019b).”

“. In contrast, oxidation of reduced sulfur compounds is nearly ubiquitous in the Chlorobiaceae (e,g. Imhoff 2014) and is likely a a synapomorphy of this clade acquired in stem lineages during Proterozoic time.”

A careful definition of “crown group” Chlorobi would be beneficial for the reader.

Response: Added. “crown group Chlorobi (i.e., the clade consisting of the last common ancestor of extant Chlorobi and all of its descendents),”

Provide refs: The evolution of photoferrotrophy in the Chlorobi appears to be a particularly

evolutionarily derived trait that evolved much more recently in Earth history than has been previously been suggested.

Response: Added

Figure 1 - the scale and bootstraps are impossible to read. Please increase the font size and resolution of the figure.

Figure 3 - same comment. This must have something to do with the PDF conversion in the PLoS system but I cannot read the labels on B. But the bootstrap font size needs to be larger.

Response: We apologize for the issues with figures. This is an ongoing issue we’ve had with PLOS’s figure upload system. We have increased the bootstrap font sizes and make sure that figure quality is preserved during re-upload. 

Reviewer #2: Ward and Shih investigate the evolutionary history of phototrophy, carbon fixation, and iron oxidation in the chlorobi, a group of organisms commonly thought to have driven the anaerobic oxidation of iron in oceans on early Earth. The results of phylogenetic analyses, comparative genomic analyses, and molecular clock analyses lead the authors to conclude that the Chlorobi are not an ancient group of anoxygenic phototrophs and were unlikely to be responsible for the deposition of BIFs. Based on the data presented, I agree with the authors, and am happy to see the work come out in deference to several heated discussions I have had with colleagues that have argued the same thing, albeit with less strong evidence. I only have several edits/comments to suggest to the authors to further improve this paper.

Line numbers would be nice.

Response: added

The abstract makes several vague claims about the role of anoxygenic photosynthesis in driving most global carbon fixation on “early earth”. This is true, but only during the latter parts of the archaeon. Prior to this, there would not have been phototrophy and it would have been chemotrophy as the primary driver. The authors need to add clarity to the time frames they are referencing. This is also true in the next sentence – do not discount the role of chemotropy – it does not hurt your arguments about the role of chlorobi in phototrophic production.

Response: these are all good points, though we left this purposely vague to avoid controversy about the ultimate age of photosynthesis (and ultimately our results suggest that Chlorobi were not present for early or late Archean time). We have rephrased things in ways that we hope will help “for much of Earth history” instead of “during the early parts of Earth history”, “early in Earth history before the rise of oxygenic photosynthesis” to “during late Archean and early Paleoproterozoic time before the rise of oxygenic photosynthesis”, changing “ancient” to “Archean and Paleoproterozoic” in the intro, etc. 

First paragraph of introduction. Most people regard the Wood Ljungdahl pathway as the most primitive CO2 fixation pathway (see papers by Martin, Russell, Boyd, and Shock). Those organisms that use the rTCA are primarily aerobes (at least those that are somewhat deeply rooted). This information was not known at the time Wachterhauser made his thoughts known.

Response: This is a good point, and we agree that rTCA is likely not as old as Wachterhauser suggested. We have added a reference to Ragsdale 2018 here as an additional example of an author positing the antiquity of rTCA more recently, demonstrating that that this continues to be a common assertion (even if we don’t think it’s well supported, as discussed later in our introduction and throughout this manuscript). 

Methods. Why not use the far more robust (albeit computationally more intensive) Clustal program to align sequences. Last line of this paragraph – remove non-essential self-citations.

Response: Citations removed. In our experience, MUSCLE performs similarly well or even better than Clustal in our use cases. This is consistent with results from from others (e.g. Pais et al., Alg. Mol Bio 2014; Pervez et al., Evo. Bioinf. 2014) who have shown that no one multiple sequence aligner performs consistently better than others across a range of metrics and datasets. 

Page 3 methods. The uniform prior of 2300-3800 for the split between Cynaobacteria and Melainabacteria is unnecessarily large. See the recent papers by Fournier for perhaps more reasonable calibration dates.

Response: The timing of the split between Cyanobacteria and Melainabacteria are extremely controversial. Fournier et al have provided calibration dates that are largely predicted based on their own molecular clock analyses with their own prior assumptions; thus, using their calibration dates impart their own biases. Given the controversy of the timing of this event, we sought the least controversial and most conservative approach to implementing the use of a uniform prior between 2300-3800, as most would agree that the split occurred within this time period, i.e., before the Great Oxygenation Event and after to the Late Heavy Bombardment. The utility of the uniform prior allows for an agnostic search within this time period for a date and range that best describes the dataset. Thus, we believe the use of our uniform prior provides a more neutral and unbiased molecular clock analysis that accurately reflects the uncertainty in this timing of this event, while not biasing the molecular clock analysis.

Page three, first paragraph results. The Chlorobi are a basal group among a clade that is sister to the other large clade. It is not necessarily sister to the other members within this clade. That it is basal is what is important to your argument.

Response: This is a subtle distinction, but we believe our phrasing is correct. The clade of Chlorobi+Kapa+Ignavi+UBA1010+Kryptonia is basal within the Bacteroidota. The clade of Chlorobi is sister to the clade of Kapa+Ignavi+UBA1010+Kryptonia. This is correct despite the fact that we are referring to sister groups of different taxonomic ranks (i.e. a single class versus a larger clade encompassing multiple classes). We have rephrased this statement to try to make that interpretation more clear: “In particular, it appears that the Chlorobi, together with a larger sister clade consisting of the classes Kapabacteria, Ignavibacteria, Kryptonia, and a class designated by GTDB as UBA10030, are basal within the broader Bacteroidota phylum.”

Page 4. Are type I RC also absent in other Bacterioidota? That seems to be critical to your argument. Also, what are the type I RC in the Chlorobi most closely related to? Finally, the idea that the RC phylogeny and the Bch synthesis protein phylogenies are incongruent means that RCs may have evolved and then were augmented by chlorophylls? Is this common in other organisms as a means of energy generation? It would be good to speculate on how this could occur based on data that we have from other organisms where this is the case (aerobic anoxygenic phototrophs, for example). The implication is RCs can function without BChl?

Response: Correct, we’ve now added “. Consistent with this scenario, Type 1 Reaction Centers are absent in other Bacteroidota.”. As stated later in this paragraph and shown in Figure 3, the reaction centers of Chlorobi are most closely related to those of Chloracidobacteria. The incongruence between the evolutionary history of reaction centers and chlorophylls is a complex topic outside the scope of this paper (but which we have tackled elsewhere, see Ward and Shih 2021). While it looks like the RCs and Bch genes in bacteria have different histories, we do think that (bacterio)chlorophylls are essential to RC-based phototrophy as we know it today, so most evolutionary scenarios invoke replacement of one component or the other associated with HGT. We have clarified this in the text: “potentially associated with the loss and replacement of older bacteriochlorophyll synthesis genes during acquisition of chlorosomes, e.g. Hanada et al., 2002, Garcia Costas et al., 2011).”

Page 4 first paragraph. You don’t acquire phototrophy via HGT but you do acquire the genes that enable phototrophy via HGT.

Response: changed to “genes for phototrophy”

Page 4 second paragraph. Your own data indicate that the rTCA cycle is not ubiquitous.

Response: We have adjusted the phrasing here to adjust and clarify this: “The rTCA cycle for carbon fixation is essentially ubiquitous in the Chlorobiaceae family” 

Page 4 second paragraph, second line to the end. This is a supporting argument at best. I would consider removing this as this phenomenon could equally be explained by expansion of niche space or variable rates of divergence/substitution.

Response: We definitely only see this as a supporting argument, and so have adjusted this to say “consistent with” rather than “supported by”

Page 5, second paragraph. Have Cyc2 genes been shown to be involved in Fe oxidation in Chlorobi. Need to provide a citation or qualify as such. Also, the use of significant in the last sentence implies that you have performed a statistical test. Use a different word here.

Response: Clarified to “Photoferrotrophic Chlorobi (but not non-phototrophic close relatives) encode a Cyc2 Cytochrome protein homologous to that putatively used in iron oxidation by members of the Zetaproteobacteria such as Mariprofundus ferrooxidans (McAllister et al. 2020), leading to the use of Cyc2 as a marker for photoferrotrophy in Chlorobi genomes although the function of this protein in iron oxidation by Chlorobi has not yet been tested in culture (Tsuji et al., 2020).” 

The accumulation of “significant” O2 at the GOE is a widely accepted phrasing (e.g. Fischer et al. 2016, Bekker et al. 2004), as the GOE is defined by oxygen reaching sufficient levels to leave detectable geochemical signatures such as the disappearance of MIF-S, the loss of redox-sensitive detrital grains, and the appearance of red beds. 

Page 6, last line of first paragraph needs a reference to authors who have suggested chlorobi are old.

Response: added. 

Conclusions. Have you ruled out photoferrotrophy in other extant lineages? Is it possible that photoferrotrophs abandoned iron when sulfide became more prevalent in oceans, etc. and thus we have no record of them. Please do a better job of trying to explain the presence of BIFs prior to the GOE here in a more ecologically appropriate manner (i.e., link changing Earth landscapes with changing selective pressures, etc.). Fe oxidation may only exist today in niche (rare) environments.

Response: We have adjusted our language slightly to account for the possibility of photoferrotrophy having been retained into extant organisms in other undescribed lineages, e.g. adding “necessarily” and “may to “Modern photo(ferro)trophs are not necessarily relicts of ancient diversity as previously presumed (e.g. Camacho et al. 2017) but may instead reflect more recent radiations following horizontal gene transfer and/or convergent evolution of metabolic traits”,

---

## [Decision Letter · Decision Letter 1]

7 Jun 2022

Phototrophy and carbon fixation in Chlorobi postdate the rise of oxygen

PONE-D-21-25104R1

Dear Dr. Ward,

We’re pleased to inform you that your manuscript has been judged scientifically suitable for publication and will be formally accepted for publication once it meets all outstanding technical requirements.

Kind regards,

Eva Elisabeth Stüeken, Ph.D.

Academic Editor

PLOS ONE

Additional Editor Comments (optional):

Dear Dr Ward,

Thanks for your submission to PlosOne! All comments have been addressed thoroughly. The manuscript is now ready for publication.

Best wishes,

Eva Stueeken

Reviewers' comments:

Reviewer's Responses to Questions

**Comments to the Author**

1. If the authors have adequately addressed your comments raised in a previous round of review and you feel that this manuscript is now acceptable for publication, you may indicate that here to bypass the “Comments to the Author” section, enter your conflict of interest statement in the “Confidential to Editor” section, and submit your "Accept" recommendation.

Reviewer #1: All comments have been addressed

Reviewer #2: All comments have been addressed

2. Is the manuscript technically sound, and do the data support the conclusions?

Reviewer #1: Yes

Reviewer #2: Yes

3. Has the statistical analysis been performed appropriately and rigorously? 

Reviewer #1: N/A

Reviewer #2: Yes

4. Have the authors made all data underlying the findings in their manuscript fully available?

Reviewer #1: Yes

Reviewer #2: Yes

5. Is the manuscript presented in an intelligible fashion and written in standard English?

Reviewer #1: Yes

Reviewer #2: Yes

6. Review Comments to the Author

Reviewer #1: My concerns and comments have been addressed in the revised submission. The authors have responded to my comments and suggestions.

Reviewer #2: (No Response)

7. PLOS authors have the option to publish the peer review history of their article (what does this mean?). If published, this will include your full peer review and any attached files.

Reviewer #1: No

Reviewer #2: No

---

## [Editor Report · Acceptance letter]

18 Jul 2022

PONE-D-21-25104R1 

Phototrophy and carbon fixation in Chlorobi postdate the rise of oxygen 

Dear Dr. Ward:

I'm pleased to inform you that your manuscript has been deemed suitable for publication in PLOS ONE. Congratulations! Your manuscript is now with our production department. 

Kind regards, 

on behalf of

Dr. Eva Elisabeth Stüeken 

Academic Editor

PLOS ONE